# Adsorption Capacity of Plastic Foils Suitable for Barrier Resuscitation

Philipp Holczmann [1], Wolfgang Lederer [1,*], Markus Isser [2], Andreas Klinger [2], Simone Jürschik [3,4], Helmut Wiesenhofer [3,4], Chris A. Mayhew [3,4] and Veronika Ruzsanyi [3,4]

[1] Department of Anaesthesiology and Critical Care Medicine, Medical University of Innsbruck, Anichstr. 35, 6020 Innsbruck, Austria
[2] Austrian Mountain Rescue Service—Tyrol, Medical Division, Florianistr. 2, 6410 Telfs, Austria
[3] Institute for Breath Research, University of Innsbruck, Innrain 66, 6020 Innsbruck, Austria
[4] Tiroler Cancer Research Institute, Innrain 66, A-6020 Innsbruck, Austria
* Correspondence: wolfgang.lederer@i-med.ac.at

**Abstract:** Chest compressions and ventilation attempts can generate aerosols during resuscitation. It is important to determine whether different materials suitable for the blanketing of cardiac arrest patients can diminish exposure to aerosols. In this study, three volatile organic compounds, ethanol, acetone, and isoprene, commonly found in human breath in moistened air, acted as substitutes for aerosols. Here, we present information on the adsorption of these volatiles to three blanketing materials: polyvinyl chloride, polyethylene, and aluminum coated polyethylene terephthalate. After exposure to the surfaces of these materials the test volatiles were quantified by the proton transfer reaction-time of flight-mass spectrometry. There was a trend towards a potentially higher reduction for acetone ($p = 0.071$) and isoprene ($p = 0.050$) on polyethylene, compared to polyvinyl chloride and aluminum coated polyethylene terephthalate during the rise interval. Adsorption capacity did not differ between the foils and was between 67% and 70%. From our studies, we propose that the aluminum-coated polyethylene terephthalate surface of space blankets prove adequate to diminish exposure to volatiles in moistened air, and hence to aerosols.

**Keywords:** device safety; medical; emergency medicine; infection transmission; personal protective equipment; rescue work; resuscitation; cardiopulmonary



## 1. Introduction

The awareness of virus transmission from aerosol-generating procedures had increased dramatically during the corona virus disease (COVID-19) pandemic [1]. Fear of infection caused delayed and low-quality basic life support (BLS) and was associated with diminished survival compared to the pre-pandemic period [2,3]. The transmission of severe acute respiratory syndrome—associated corona virus (SARS-CoV-2)—was reported to arise from respiratory secretions and from contaminated surfaces [1,4,5]. The emission rates of aerosols are particularly high during speaking, singing, and exercises [6,7]. Airborne particles, especially tiny aerosols, can remain suspended in the environmental air for prolonged periods thus creating persistent hazards to the health of rescuers and patients [5,8,9]. There is evidence that chest compressions and ventilation have the potential to generate aerosols [10–12]. The lack of personal protective equipment (PPE) predominantly affects lay rescuers performing cardiopulmonary resuscitation (CPR) in the prehospital setting [13,14]. Currently, good practice statements and consensus on recommendations from resuscitation councils suggest that lay rescuers consider compression-only CPR and public-access defibrillation (PAD) during the COVID-19 pandemic [9,12,15–17].

Until the COVID-19 pandemic, CPR barrier devices such as pocket masks and face shields were regarded sufficient for lay rescuers to avoid infection from contact to the mucous membranes of the patient [18]. For professional rescuers, the use of PPE had

become mandatory [19]. However, even with full PPE, the risk of transmission cannot be completely eliminated [20]. Whenever rapid tracheal intubation appears not practicable, ventilation by a bag-valve-mask (BVM) device attached to a high-efficiency particulate absorbing (HEPA) filter was recommended by international guidelines [12,17]. Despite the use of HEPA filters, air leakage from BVM ventilation can expose rescuers to virus-loaded aerosols. Recent studies advocate that instead of covering only the face of the victim it may be advisable to completely cover the contact area of the victim's head and chest with plastic shields [21,22].

In this experimental study we used volatiles in moistened air as substitutes for aerosols. The aim is to estimate the potential of adsorption on different plastic foils suitable as a thin foil body-shield resuscitation barrier device to protect from infection.

## 2. Factors Influencing Adhesion on Plastic Foils

### 2.1. Aerosols

Regarding the aerogenic transmission of the SARS-CoV-2-virus, we have to consider that the emission rates of aerosols during breathing may exceed 1000 particles per second [23].

The fluid component of the pathogen-containing particles from exhaled aerosols evaporates within fractions of a second, and in particular when aerosols are smaller than 1 μm [23]. Furthermore, the spatial distribution and the half-life of aerosols on surfaces are dependent on several factors of the surrounding air, most of all temperature, humidity, and dilution ratio [23,24]. In an unpublished preliminary study, we investigated aerosols made from supersaturated vapor condensed onto ammonium nitrate commonly using a stationary condensation particle counter (Advanced CPC 5.416 Grimm Aerosol Technik, Ainring, Germany). We established a test bag 60 cm × 60 cm in size (exposed area: 14,400 $cm^2$) made of different plastic foils. Measurements were performed at 10 s and 60 s after inflation of 2500 mL of aerosol containing air over 1 min. From these preliminary investigations, we concluded that the results strongly depend on the kind of selected aerosol, and that at least one quarter of reduction in aerosol content resulted from evaporation. We therefore have used in this study volatiles commonly found in human breath [25,26] in moistened air, to mimic the behavior of aerosols on different plastic foils.

### 2.2. Volatile Organic Compounds (VOCs)

Acetone, isoprene, and ethanol, concentrations in breath vary considerably (both inter-individually and intra-individually), but typical average volume mixing ratios are of the order of 500 $ppb_v$ for acetone, 130 $ppb_v$ for isoprene, and 80 $ppb_v$ for ethanol [26,27]. Importantly, these three selected volatiles differ in respect to their water-solubility: ethanol (high), acetone (moderate), and isoprene (low). There are several confounding factors for both real emergency scenarios and the experimental setting, including form and spatial dimension of the test materials, relative humidity, room temperature, length of exposure, surface energy, and surface texture of foils, to mention a few. We investigated volatiles in the understanding that in moistened air they can substitute for aerosols.

## 3. Materials and Methods

We investigated surface properties of three different materials suitable for blanketing patients during CPR. We evaluated the adsorption of the volatiles in moistened air to the surfaces of polyvinyl chloride (PVC), polyethylene (PE), and aluminum coated polyethylene terephthalate (Al-PET), using proton transfer reaction-time of flight-mass spectrometry (PTR-ToF-MS) [28,29]. The results are discussed in terms of their impact on barrier resuscitation attempts in the out-of-hospital setting. Study design and manuscript presentation followed the SQUIRE 2.0 Revised Standards for Quality Improvement Reporting Excellence to improve the quality, safety, and value of healthcare [30]. As the experimental design was focusing on the physical properties of the materials, and not on human beings, the institutional guidelines of the Clinical Trial Center (CTC) exempted the study from the need

of an ethical approval. Investigations were conducted at the Institute for Breath Research, University of Innsbruck, Austria.

### 3.1. Barrier Materials and Test Volatiles

In this experimental design, the adsorption of the volatiles was quantified using foil segments of 100 cm × 24 cm (2400 cm$^2$). This area corresponds to a sixth (16.7%) of a 120 cm × 120 cm (14,400 cm$^2$) foil that would be large enough to cover the head and chest of a patient's body. We focused our studies on foils that have an increased chance of being available in an out-of-hospital setting, namely three foils commonly used in construction, operation theatre and emergency medicine:

(1) Plasticized PVC sheets (d50 universalpresenning; JUFOL GmbH, D-86154, Augsburg, Germany). These foils are transparent, smooth, i.e., without structured surface, with dimensions of 4000 cm × 5000 cm and a thickness of 50 μm.

(2) PE sheets (Operation drapes, Henry Schein Inc., Melville, NY, USA). These green colored foils are semi-transparent with a rough and structured surface, dimensions of 120 cm × 120 cm and a thickness of 70 μm.

(3) Al-PET sheets (First Aid Blanket; M06299; Franz Kalff GmbH, Euskirchen, Germany). These foils are also semi-transparent, with either gold or silver (1% aluminum coated) color. They are robust, light, low-weight, low-bulk, dimensions of 160 cm × 210 cm, and a thickness of 12 μm.

Ethanol (C$_2$H$_6$O; CAS-Nr.: 64-17-5), isoprene (C$_5$H$_8$; CAS-Nr.: 78-79-5) and acetone (C$_3$H$_6$O; CAS-Nr: 67-64-1) were purchased from Sigma-Aldrich Chemie GmbH, Munich, Germany, with stated purities of 99.9% for acetone, 99.5% for isoprene and 96% for ethanol. These chemicals were used without further purification.

To mimic concentrations in breath, we established a test gas by adding 0.5 μL ethanol, 3.5 μL acetone and 1.5 μL isoprene into a 1 L glass bulb (Supelco$^®$ Analytical, Vienna, Austria) that had been evacuated at a temperature of 65 °C overnight in an incubator (Memmert GmbH + Co.KG, Schwabach, Germany). Once the volatiles were introduced, the pressure in the bulb was equalized to barometric pressure using purified nitrogen gas (99.9999% purity).

### 3.2. Measurement Procedures

The investigations were performed using a glass cylinder of length 1 m and with a 7.5 cm inner diameter. The cylinder has two lateral openings, each of 3.5 cm diameter, separated by 77 cm (Figure 1). The inner surface was covered with either PVC, PE, or Al-PET foil segments (exposed area: 2360 cm$^2$; volume: 4400 mL).

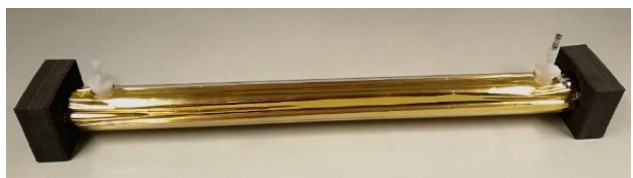

**Figure 1.** Measurement cylinder consisting of a glass tube covered, in this case, with Al-PET foil on the inner surface. It is plugged with two closing caps on the ends and has two lateral openings, one for introducing the test gas and the other connected to the inlet of the PTR-ToF-MS for sample analysis.

After the installation of one of the three barrier materials, the cylinder was rinsed with filtered air at a flow rate of 70 mL/min for 24 h in order to remove possible adsorbed volatiles from the surfaces. Blank values were verified by PTR-ToF-MS measurements before each test run. As humidity is very high in exhaled air, we did not evaluate the influence of moderate and low percentages of relative humidity on adsorption. Purified air was moistened by passing it through a water reservoir, thereby providing a buffer gas exceeding ~80% relative humidity. This was used to passivate the glass cylinder for 30 min prior to and

also during the humidified VOC studies with a flow rate of 250 mL/min. The experiments were carried out at room temperature, which was maintained at 21 °C ± 1 °C according to fluctuations from the air-conditioning (range: 20 to 23 °C). The room temperature was measured and recorded prior to each experiment.

1.6 mL of the test gas containing a volatile were drawn from the gas bulb and injected via the inlet septum (Thermogreen, Merck KGaA, Darmstadt, Germany) into the glass cylinder using gas-tight syringes (Hamilton, Merck KGaA, Darmstadt, Germany). After a 45 s waiting time (by stopping the inlet flow for a total of 70 s to receive an adequate distribution of the compounds in the glass cylinder) the PTR-ToF MS data acquisition was started. Then, the humidified air flow was restarted after the outlet of the glass cylinder had been directly coupled via a 1/16 inch peak tubing to the heated inlet of the PTR-ToF-MS. The inlet sample flow of the PTR-ToF-MS was adjusted to 250 mL/min, thereby avoiding pressure differences in the glass cylinder during measurements. Data were taken on-line in real time over approximately 30 min until the wash-out had achieved volatile concentrations close to the baseline level.

### 3.3. Analytical Measuring Device and Data Acquisition Parameters for PTR-ToF-MS

Proton Transfer Reaction—Time of Flight—Mass Spectrometry (PTR-ToF-MS) (a PTR 6000 X2, Ionicon Analytik GmbH, Innsbruck, Austria) was used in our study [31,32]. This instrument allows near to real-time detection (0.1 s temporal resolution) with fast testing cycles of approximately 1 s and high sensitivity of test volatiles after exposure to different surfaces taking into account temperature, pressure, buffer gas flow, and ion formation from the sample [27,28]. The instrument consists of a hollow-cathode ion source producing the $H_3O^+$ ions, which is followed by a drift tube where proton transfer reactions between the sample gas and the hydronium ions occur. After the reagent and product ions entered the analyzer, guided by a hexapole, a time-of-flight mass spectrometer separates ions according to their time-of-flights, which are converted a mass-to-charge-ratios ($m/z$) through use of a mass calibration. The drift tube was maintained at a constant pressure of 2.6 mbar and a temperature of 80 °C. All experiments were carried out at a reduced electric field of 124 Townsend (Td) [28].

Data were processed using the PTR 6000 X2 internal software (PTR-MS Viewer 3.4.3.12). The reagent and product ion mass spectral peaks for acetone, isoprene and ethanol were fitted to pseudo Voigt profiles, using the Ionicon Analytik GmbH software (version 4.0), from which the peaks' positions and their areas were obtained. Peak positions used were for protonated acetone ($C_3H_7O^+$) $m/z$ 59.050, isoprene ($C_5H_9^+$) $m/z$ 69.070, and ethanol ($C_2H_7O^+$) $m/z$ 47.050 (including its water cluster ($C_2H_7O^+.H_2O$) $m/z$ 65.060). Since the signals for the reagent ions $H_3{}^{16}O^+$ and $H_3{}^{16}O^+.H_2O$ at $m/z$ values of 19.018 and 37.029, respectively, were saturated, their isotopes $H_3{}^{18}O^+$, $m/z$ 21.023, and $H_5{}^{16}O^{18}O^+$, $m/z$ 39.033, were used to get normalized ion signals in order to take into account the differences in the water content of the samples. An average of the three independent measurements were used to obtain the wash out characteristics of the product ions resulting from the investigated compounds.

### 3.4. Calculations of Reduction and Adsorption Capacity and Statistical Analysis

We assumed an equal distribution of the volatiles in the glass cylinder when peak concentrations (PC) have been achieved in the samples drawn from the cylinder. In order to simplify dynamic responses a constant dilution with moistened air at a flow rate of 250 mL/min was established after an initial injection of the test gas. In order to calculate different area segments at half peak concentration ($PC_{50}$), we distinguished between the following intervals: half peak concentration interval ($PCI_{50}$), rise interval (RI), decline interval (DI) and estimated dilution interval (EDI) after peak concentration. This is illustrated in Figure 2 for an acetone and polyvinylchloride foil measurement.

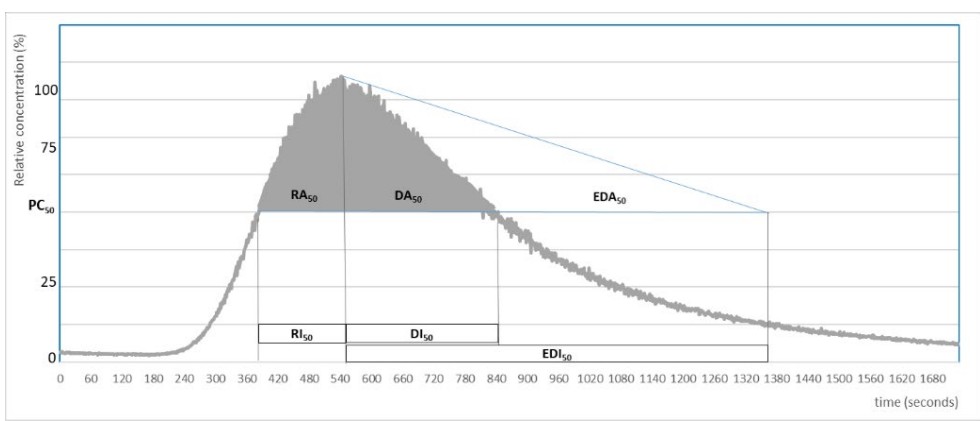

**Figure 2.** Adsorption capacity of acetone on polyvinylchloride foils. Areas at 50% peak concentration ($PC_{50}$) during rise ($RA_{50}$), during decline ($DA_{50}$) and after estimated 50% dilution ($EDA_{50}$). Intervals between 50% proof and peak maximum during rise ($RI_{50}$), between peak maximum and 50% proof during decline ($DI_{50}$) and between peak maximum and estimated 50% dilution ($EDI_{50}$).

The area-under-the-curve (AUC) of a test substance relative concentration versus time was calculated with OriginLab©(Origin Pro 217G) and used to quantify the wash-out rate [33]. We distinguished between areas at 50% proof ($RA_{50}$) during rise, areas at 50% proof ($DA_{50}$) during decline and areas at 50% proof ($EDA_{50}$) during estimated dilution. Areas between 50% concentration ($PC_{50}$) and peak concentration during rise interval ($RI_{50}$), and between peak concentration and 50% concentration during decline interval ($DI_{50}$) and estimated dilution interval ($EDI_{50}$) after peak concentration required for the VOC concentration to decrease by 50% were differentiated (as shown in Figure 2 for acetone). As the overall reduction (R) of test volatiles during rise was proposed to arise from dispersion, dilution, and adsorption, we calculated the ratio between $RA_{50}$ (antecedent) and $DA_{50}$ (consequent) at 50% proof of the test volatiles. The smaller the rise/decline ratio, the higher the anticipated drop of test substances during RI (1).

$$R = RA_{50}/DA_{50} \tag{1}$$

We used a linear course of dilution with moistened air at a constant flow after peak concentration until the concentration had decreased to 50%. 50% reduction by dilution of test gas in the cylinder between the two lateral openings was anticipated 900 s after PC. $DA_{50}$ was calculated using the area of a right-angled triangle with half of the product of the two legs $DI_{50}$ and $PCI_{50}$ of a right triangle (2).

$$DA_{50} = (DI_{50} \times PCI_{50})/2 \tag{2}$$

Similar $EDA_{50}$ was calculated using half of the product of $EDI_{50}$ and $PCI_{50}$ (3).

$$EDA_{50} = (EDI_{50} \times PCI_{50})/2 \tag{3}$$

Adsorption capacity (A) of a foil after the maximum proof was reached was determined from the ratio (quotient) of the $DA_{50}$ (antecedent) and the estimated $EDA_{50}$ (consequent) at 50% proof of the test volatiles (4).

$$A = DA_{50}/EDA_{50} \tag{4}$$

When reducing the fraction, the ratio of the areas $DA_{50}$ and $EDA_{50}$ corresponded with the ratio of intervals $DI_{50}$ (dividend) and $EDI_{50}$ (divisor).

$$A = ((DI_{50} \times PCI_{50})/2))/((EDI_{50} \times PCI_{50}/2)) \tag{5}$$

$$A = DI_{50}/EDI_{50} \tag{6}$$

Consequently, the values of A(%) represent the percentage of adsorption capacity after peak concentration (7).

$$A(\%) = (1 - DI_{50}/EDI_{50}) \times 100 \tag{7}$$

Descriptive statistics were applied using statistical package for the social sciences from International Business Machines Corporation (IBM SPSS Statistics Standard 26, Armonk, NY, USA) to determine measures of central tendency (median) and measures of dispersion (range, standard deviation, variance, minimum and maximum). Exponential rise and decline as well as AUC were calculated using OriginLab© (OriginLab Corporation, Northampton, MA, USA). Data were screened for normal distribution with the Shapiro-Wilk test. The non-parametric Kruskal-Wallis test was used for comparison of outcomes among the three foils by estimating the differences in ranks. Power calculations have shown only small standard deviations are to be expected in the measurements. Therefore, three runs per foil were deemed sufficient to achieve an adequate sample size using an online sample size calculator [34].

## 4. Results and Discussion

### 4.1. Process Measures and Outcome

In a total of nine runs (three foils and three test volatiles; with tests repeated three times), the time to reach the peak maximum was found to be similar for acetone, isoprene, and ethanol, after dispersion on the surfaces of PVC, PE, and Al-PET. Standard deviations were high, particularly during RI compared to DI. This might result from small fluctuations in room temperature. However, we observed little variation within the data between differences in the solubility and volatility of the three volatiles (Table 1). This underlines the accuracy of the data in terms of validity and reliability.

**Table 1.** Time intervals including time to peak (T to peak), half-life rise (T 1/2 rise), half-life decline (T 1/2 decl.) after exposure of acetone, isoprene, ethanol to the surfaces of polyvinyl chloride (PVC), polyethylene (PE) and aluminum-coated polyethylene terephthalate (Al-PET).

| *Characteristics* | **PVC** | **PE** | **Al-PET** | *p*-**Value** |
|---|---|---|---|---|
| **Acetone** | m ± SD | m ± SD | m ± SD | - |
| T to peak (s) | 547.0 ± 62.2 | 587.3 ± 84.5 | 692.0 ± 102.6 | 0.721 |
| T 1/2 rise (s) | 444.0 ± 31.1 | 451.7 ± 69.0 | 524.3 ± 72.1 | 0.543 |
| T 1/2 decl. (s) | 287.7 ± 52.5 | 306.7 ± 9.7 | 295.7 ± 27.6 | 0.879 |
| **Isoprene** | - | - | - | - |
| T to peak (s) | 593.0 ± 84.5 | 583.7 ± 89.5 | 691.0 ± 103.3 | 0.721 |
| T 1/2 rise (s) | 444.7 ± 33.7 | 452.7 ± 67.8 | 525.0 ± 70.2 | 0.543 |
| T 1/2 decl. (s) | 292.7 ± 45.7 | 291.7 ± 12.7 | 286.3 ± 31.2 | 0.993 |
| **Ethanol** | - | - | - | - |
| T to peak (s) | 596.7 ± 39.4 | 587.3 ± 84.5 | 708.3 ± 92.6 | 0.418 |
| T 1/2 rise (s) | 438.7 ± 29.0 | 439.0 ± 61.3 | 497.7 ± 54.0 | 0.629 |
| T 1/2 decl. (s) | 279.7 ± 39.2 | 270.0 ± 18.8 | 272.3 ± 37.6 | 0.929 |

Overall, the half-life of a test volatile did not differ significantly among the foils (Table 1), although the ratio between rise interval and decline interval at 50% proof was lower *in trend* with acetone ($p = 0.071$) and isoprene ($p = 0.050$) on PE compared to PVC and Al-PET (Table 2).

**Table 2.** Ratio between the area at 50% proof during rise interval and the area at 50% proof during decline interval for acetone, isoprene, ethanol adsorbed on the surfaces of polyvinyl chloride (PVC), polyethylene (PE) and aluminum-coated polyethylene terephthalate (Al-PET).

| *Characteristics* | **PVC** | **PE** | **Al-PET** | *p*-**Value** |
|---|---|---|---|---|
| - | m $\pm$ SD | m $\pm$ SD | m $\pm$ SD | - |
| Acetone | 0.58 $\pm$ 0.06 | 0.43 $\pm$ 0.03 | 0.59 $\pm$ 0.1 | 0.071 |
| Isoprene | 0.53 $\pm$ 0.01 | 0.40 $\pm$ 0.03 | 0.60 $\pm$ 0.11 | 0.050 |
| Ethanol | 0.48 $\pm$ 0.02 | 0.52 $\pm$ 0.11 | 0.67 $\pm$ 0.07 | 0.232 |

This applied in particular to the test volatiles with moderate (acetone) and low (isoprene) water solubility. Overall, reduction during RI was highest for isoprene on PE (60%), whereas adsorption capacity during DI did not differ among the foils (Table 3). Adsorption capacity of Al-PET foils was 67% for acetone, 68% for isoprene and 70% for ethanol, respectively (Table 3). Space blankets made of Al-PET proved adequate potential to diminish exposure to substances comparable to polyvinyl chloride.

**Table 3.** Percentages of overall reduction during rise interval and of adsorption capacity during decline interval of acetone, isoprene, ethanol on the surfaces of polyvinyl chloride (PVC), polyethylene (PE) and aluminum-coated polyethylene terephthalate (Al-PET).

| - | **PVC** | **PE** | **Al-PET** |
|---|---|---|---|
| *Overall reduction during rise interval* | | | |
| Acetone (%) | 42 | 57 | 41 |
| Isoprene (%) | 47 | 60 | 40 |
| Ethanol (%) | 52 | 48 | 33 |
| *Adhesion capacity during decline interval* | | | |
| Acetone (%) | 68 | 66 | 67 |
| Isoprene (%) | 67 | 68 | 68 |
| Ethanol (%) | 69 | 70 | 70 |

*4.2. Relevance to Rationale and Specific Aims*

This study is relevant for lay rescuers to diminish their risk of contracting air-borne infections during CPR. There is evidence that CPR, including compression-only CPR, has the potential to generate aerosols [10–12]. Paroya et al. detected aerosols on the gloves and gowns of both the BVM provider and the chest compression provider [11]. As space blankets are basic components of first aid kits, they are readily available to lay rescuers and hence can be used to remove aerosols from air leakage by adsorption and drainage below the barrier. However, further clinical studies are required to quantify how the blanketing of a patient with a plastic foil can protect against contamination by blood, vomit, and secretions, and whether it can prompt an even greater willingness of lay rescuers to start resuscitation [35].

## 5. Implications

*5.1. Association between Intervention and Outcome*

In this experimental study we focused on the adsorption of volatiles in humid air, as substitutes of aerosols, to the surfaces of plastic foils. The reduction of test volatiles during initial exposure is proposed to arise from dispersion, dilution, and adsorption [36]. There are certain limitations that need to be considered when interpreting the results. Volatiles cannot completely substitute aerosols. Dispersion and consequently adsorption may differ considerably. Presumably, the half-life intervals of the test volatiles may have been also influenced by differences in water solubility. In addition, the varying surface properties of the foils related to different hydrophobicity could contribute to differences in adsorption [37]. Under laboratory conditions, we have compared the characteristics of three selected volatiles adsorbed on the surface of three commonly used foils. The

concentrations of selected volatiles in the test gas resembled physiologic conditions of these volatiles in breath. As a consequence of the small test gas volume in relation to the large surface of foil fragments, we assumed that interactions between different volatiles should have been negligible. However, we did not test different mixtures of ethanol, isoprene, and acetone, and cannot tell whether adsorption capacity is influenced by the altered concentrations of substances. In addition, the flow in the measurement cylinder may not represent the flow conditions between patient and foils in real-life situations. Thus, the limited representativeness of data and the generalizability of findings have to be taken into account. However, we did not observe marked differences in the adsorption of the three volatiles investigated between the Al-PET surface and the PVC surface that account for space blanket surfaces. Conceivably, there are different modes of action. First, foils as mechanical barriers between rescuer and patient can protect against pathogens transmitted by skin contact, aerosol, or droplet infection, by drainage and dilution below the foil [35]. Second, the adsorption of pathogen loaded aerosols to the surface of plastic foils further diminishes the risk of transmission. Third, additional protective effects may arise from the antimicrobial properties of the surface, e.g., survival of human coronaviruses was reported to be significantly shorter on aluminum surfaces than on PVC surfaces [36].

### 5.2. Impact on People and System (PAD)

Increasing safety and motivation in lay rescuers is vital as bystander CPR and the public location of emergency were reported as independent prognostic factors for survival to hospital discharge in OHCA patients [38].

### 6. Conclusions

We used volatiles in humid air to mimic the behavior of aerosols interacting with plastic foils. Our findings revealed that there is a trend towards a potentially higher reduction for acetone and isoprene on polyethylene, compared to polyvinyl chloride and aluminum-coated polyethylene terephthalate during the rise phase of the measurements. The adsorption capacity of aluminum-coated polyethylene terephthalate foils was similar to polyethylene and polyvinyl chloride, indicating that the aluminum-coated polyethylene terephthalate surface of space blankets is sufficiently adequate to diminish exposure to volatiles in moistened air, and hence to aerosols.

**Author Contributions:** Conceptualization, M.I., W.L. and C.A.M.; methodology A.K., S.J., H.W. and V.R.; validation, P.H., W.L., S.J., V.R. and C.A.M.; formal analysis, P.H., A.K., S.J. and V.R.; literature research, P.H., W.L., A.K. and C.A.M.; writing—original draft preparation, W.L., M.I., S.J. and V.R.; writing—review and editing, P.H., W.L., S.J., H.W., V.R. and C.A.M.; visualization, P.H., A.K. and M.I.; supervision, C.A.M.; project administration, M.I. and H.W. All authors have read and agreed to the published version of the manuscript.

**Funding:** This research received no external funding.

**Institutional Review Board Statement:** Not applicable.

**Informed Consent Statement:** Not applicable.

**Data Availability Statement:** Not applicable.

**Conflicts of Interest:** The authors declare no conflict of interest.

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
