# Peer review of "Adsorption Capacity of Plastic Foils Suitable for Barrier Resuscitation"

_coatings, doi:10.3390/coatings12101545_

Round 1

Reviewer 1 Report

Holczmann et al. have carried out an experimental and statistical study in order to determine the adsorption capacity of 3 materials used as blanketing during resuscitation.

Authors have described the analytical and statistical method in order to determine the adsorption capacity of three small VOC on the surface of three materials.

The paper is overall good written and the objective of the study is clearly defined.

Here some interrogations to complete the data.

Authors described that acetone, ethanol and isoprene as markers in moistened air.

·       Did authors analyze blanks with only blankets?

·       The measurements have been carried out offline or online? Is  not clear in the text.

·       Did authors carried out mix conditions of ethanol, isoprene and acetone in order to see if there any differentiation of adsorption capacity of materials in the presence of other molecules?

·       The humidity was tested at one percentage, did author tested other conditions of humidity?

·       What about the temperature, at which temperature the study has been carried out and did authors measured the temperature of the study?

Author Response

We thank the reviewer for the conscientious revision and provide a point-by-point response. In addition, we performed minor corrections of spelling and syntax and described methods more clearly.   

Authors described that acetone, ethanol and isoprene as markers in moistened air.

  • Did authors analyze blanks with only blankets?

We specified lines 139-140:

Blank values were verified by PTR-ToF-MS measurements before each test run.

  • The measurements have been carried out offline or online? Is not clear in the text.

We specified in line 154:

Data were taken on-line in real time over approximately 30 minutes …

  • Did authors carried out mix conditions of ethanol, isoprene and acetone in order to see if there any differentiation of adsorption capacity of materials in the presence of other molecules?

We specified in limitations lines 308 -312:

As a consequence of the small test gas volume in relation to the large surface of foil fragments we assumed that interactions between different volatiles should have been negligible. However, we did not test different mixtures of ethanol, isoprene and acetone and cannot tell whether adsorption capacity is influenced by altered concentrations of substances.

  • The humidity was tested at one percentage, did author tested other conditions of humidity?

We specified in methods lines 140 -141:

As humidity is very high in exhaled air we did not evaluate the influence of moderate and low percentages of relative humidity on adsorption.

  • What about the temperature, at which temperature the study has been carried out and did authors measured the temperature of the study?

We specified in lines 145 – 147:

The experiments were carried out at room temperature, which was maintained at 21 °C + 1 °C according to fluctuations from air-conditioning (range: 20 to 23 °C). The room temperature was measured and recorded prior to each experiment.

Reviewer 2 Report

This is generally a well-written and comprehensive article regarding the potential of adhesion on different plastic foils suitable as thin foil body-shield resuscitation barrier devices to protect from infection. I consider that the work is well-structured, and the results are presented properly. The article is valuable and sound, with important clinical implications, but I suggest a more detailed presentation of the limitations of the study.

Author Response

This is generally a well-written and comprehensive article regarding the potential of adhesion on different plastic foils suitable as thin foil body-shield resuscitation barrier devices to protect from infection. I consider that the work is well-structured, and the results are presented properly. The article is valuable and sound, with important clinical implications, but I suggest a more detailed presentation of the limitations of the study.

We thank the reviewer for this important comment and provide  more detailed  information in limitations. In addition, we performed minor corrections of spelling and syntax and described methods more clearly.  

We specified in lines 299-302:

There are certain limitations that need to be considered when interpreting the results. Volatiles cannot completely substitute aerosols. Dispersion and consequently adsorption may differ considerably.

and in lines 305 - 314:

Under laboratory conditions, we have compared the characteristics of three selected volatiles adsorbed on the surface of three commonly used foils. Concentrations of selected volatiles in the test gas resembled physiologic conditions of these volatiles in breath. As a consequence of the small test gas volume in relation to the large surface of foil fragments we assumed that interactions between different volatiles should have been negligible. However, we did not test different mixtures of ethanol, isoprene and acetone and cannot tell whether adsorption capacity is influenced by altered concentrations of substances. In addition, the flow in the measurement cylinder may not represent the flow conditions between patient and foils in real-life situations. Thus, limited representativeness of data and generalizability of findings have to be taken into account.

Reviewer 3 Report

The work presented by Lederer, and coworkers titled “Adsorption Capacity of Plastic Foils Suitable for Barrier Resuscitation” is written well and explained systematically. Before publishing this article, I have some questions.

Comment 1. “volatile organic compounds, ethanol, acetone, and isoprene, commonly found in human breath” can the author provide some reference related to it in the main text?

Comment 2. The citation is not complete few citation such as https://www.nature.com/articles/s43856-022-00103-w, https://doi.org/10.1098/rsfs.2021.0078, https://doi.org/10.1073/pnas.2202521119 should be cited.

Comment 3. In table 1 the standard deviation was quite high. Can author explain the reason behind this.

Comment 4. The cited references are not appropriate in reference no 2, 4, 5 and 6. Why underline?

Author Response

We thank the reviewer for the valuable comments and the helpful suggestions and provide a point-by-point response. In addition, we performed minor corrections of spelling and syntax and described methods more clearly.   

The work presented by Lederer, and coworkers titled “Adsorption Capacity of Plastic Foils Suitable for Barrier Resuscitation” is written well and explained systematically. Before publishing this article, I have some questions.

Comment 1. “volatile organic compounds, ethanol, acetone, and isoprene, commonly found in human breath” can the author provide some reference related to it in the main text?

We provided 2 more references as requested

King et al. Isoprene and acetone concentration profiles during exercise on an ergometer. J Breath Res. 2009, 3(2), 027006.

Diskin, A.M.; Spanel, P.; Smith, D. Time variation of ammonia, acetone, isoprene and ethanol in breath: a quantitative SIFT-MS study over 30 days. Physiol Meas. 2003, 24(1), 107-119.

Comment 2. The citation is not complete few citation such as https://www.nature.com/articles/s43856-022-00103-w, https://doi.org/10.1098/rsfs.2021.0078, https://doi.org/10.1073/pnas.2202521119 should be cited.

We added 2 of the suggested references on particle emission:

Orton et al. A comparison of respiratory particle emission rates at rest and while speaking or exercising. Commun Med (Lond). 2022, 2,44.

Mutsch et al. Aerosol particle emission increases exponentially above moderate exercise intensity resulting in superemission during maximal exercise. Proc Natl Acad Sci U S A. 2022, 119(22), e2202521119.

and 2 more references on methods:

Malásková et al. , Compendium of the Reactions of H3O+ With Selected Ketones of Relevance to Breath Analysis Using Proton Transfer Reaction Mass Spectrometry. Front Chem. 2019, 7, 401.

Weiss et al. A Selective Reagent Ion-Time-of-Flight-Mass Spectrometric Study of the Reactions of O2+• with Several Volatile Halogenated Inhalation Anaesthetics: potential for breath analysis. Eur. Phys. J. D 2022, In press

Comment 3. In table 1 the standard deviation was quite high. Can author explain the reason behind this.

We specified in results lines 259-261:

Standard deviations were high, particularly during RI compared to DI. This might result from small fluctuations in room temperature.

Comment 4. The cited references are not appropriate in reference no 2, 4, 5 and 6. Why underline?

We regret the mistake and removed the lines as requested